# Carbon and Nitrogen Dynamics Affected by Drip Irrigation Methods and Fertilization Practices in a Pomegranate Orchard

**Rebecca Tirado-Corbalá [1],\***, **Suduan Gao [2]**, **James E. Ayars [2]**, **Dong Wang [2]**, **Claude J. Phene [3] and Rebecca C. Phene [4]**

1   Agro-Environmental Sciences Department, University of Puerto Rico-Mayagüez, P.O. Box 9000, Mayagüez, PR 00681, USA
2   USDA, Agricultural Research Service, San Joaquin Valley Agricultural Sciences Center, Parlier, CA 93648, USA; suduan.gao@ars.usda.gov (S.G.); james.ayars@gmail.com (J.E.A.); dong.wang@ars.usda.gov (D.W.)
3   SDI+, P.O. Box 314, Clovis, CA 93613, USA; claudejphene@gmail.com
4   Kearney Agricultural Research and Extension Center, University of California, 9240 S. Riverbend Ave., Parlier, CA 93648, USA; rcphene@ucanr.edu
\*   Correspondence: rebecca.tirado@upr.edu or rebeccatiradocorbala@gmail.com; Tel.: +1-787-370-9179

**Abstract:** Knowledge of carbon (C) and nitrogen (N) dynamics under different irrigation practices in pomegranate orchards is novel and essential to develop sustainable production systems. The aim of this research was to determine the effect of high-frequency drip irrigation and different rates of N fertilizer on C and N distribution in the soil and N uptake by pomegranate fruit and leaves. The main treatments were surface drip irrigation (DI) and subsurface drip irrigation (SDI), and the sub-treatments used were three initial N rates (N1, N2, and N3). As trees grew larger, the N application rate increased. From 2013–2015, trees received the following rates of N: 62–113 (N1), 166–263 (N2), or 244–342 kg/ha (N3). Soil and leaf total C (TC) and N (TN), soil dissolved organic C (DOC), soil nitrate ($NO_3^-$), and total N uptake by fruit were evaluated between 2012 and 2015. Soil samples were collected to 120 cm depth at 15 cm increments. DI resulted in higher concentrations of TN, TC, $NO_3^-$, and DOC in the upper 75 cm depth than SDI. The N3 treatment resulted in higher concentrations of TN, TC, $NO_3^-$, and DOC under both DI and SDI. Neither DI nor SDI at the N1 or N2 levels increased TN and $NO_3^-$ concentrations at 105–120 cm soil depth, indicating reduced leaching risk using high-frequency drip irrigation. Higher N uptake by fruit was observed in SDI than in DI in 2014 and 2015, and in N2 and N3 treatments compared with N1 in 2013 and 2014. The data indicate that the application rate at 166–263 kg/ha (N2) provided sufficient N for a 4–6-year-old pomegranate orchard and that high-frequency SDI is a promising technology for achieving higher N use efficiency and minimizing leaching loss of $NO_3^-$ and DOC.

**Keywords:** dissolved organic carbon; total carbon; total nitrogen; nitrate; subsurface drip irrigation; N uptake

## 1. Introduction

Pomegranate (*Punica granatum* L.) is a deciduous fruit tree native to central Asia [1]. Its production in the United States has increased in recent years due to consumer interest in pomegranate organoleptic characteristics and beneficial effects on human health [2]. California is the top pomegranate production state in the U.S. with approximately \$47 billion production in 2015 [3,4], >99% of all production in the United States, with 783 farms occupying a total of 13,041 ha primarily located in the Central Valley [4]. Drought has been increasingly challenging to crop production in California, and pomegranate has been

identified as a promising specialty crop due to its high drought and salinity tolerance [5–8]. Despite being an "ancient" crop, it has been studied primarily for medicinal uses [9,10] and little is known about the water and fertilizer requirements of a mature pomegranate orchard, especially under different drip irrigation methods. Water and fertilizer research on pomegranate has been conducted primarily in Mediterranean climates using furrow [11] or flood irrigation systems [12], with some drip irrigation studies in Western Asia (i.e., Fars Province, Iran) under arid and semi-arid conditions [13–15]. Several pomegranate growers in California irrigate the crop with surface drip irrigation (DI) systems [4,16] with little knowledge of the actual water and fertilizer requirements.

Agriculture is one of the largest drivers of spatiotemporal changes in global N and C cycles. Under a Mediterranean climate, 100% of the crop production in the Central Valley of California is irrigated with an annual output exceeding $30 billion [3], and approximately 40%–80% of the total water supply is used for this irrigation [17,18]. High levels of nitrate ($NO_3^-$) in the groundwater represent a major problem in California aquifers [19,20], and irrigated agriculture is a significant source of $NO_3^-$ pollution [21,22]. Synthetic N fertilizers (54%), animal manure (33%), and irrigation water (8%) represent the three major sources of $NO_3^-$ found in California groundwater [20]. Export of DOC and $NO_3^-$ from surface to groundwater is strongly influenced by agricultural practices [19,23]. Chomycia et al. [24] found that water from tile drains under manure-irrigated fields, which ultimately flows into the San Joaquin River in California, contained DOC concentrations about three orders of magnitude higher than in the San Joaquin River. A better understanding of the effects of irrigation and fertilization practices on pomegranate field DOC and $NO_3^-$ mobility will assist in the development of more environmentally sound management strategies.

Groundwater is the major source of drinking water for many communities in the Central Valley of California. The 2014 Groundwater Act (Sustainable Groundwater Management Act) of California made protecting groundwater quality a priority for public health by reducing N leaching in basins that adopt salt nutrient management. N that is applied to crops but not removed by harvest or lost via air emission and runoff is subject to leaching from the root zone to groundwater [25]. Options for reducing $NO_3^-$ pollution have been identified for all sources. For croplands, where less than 40% of applied N is removed by crop harvest, several management practices have been suggested to reduce $NO_3^-$ leaching to groundwater. Harter et al. [21] used efficient irrigation design and fertilization management to increase N use efficiency. Studies done by Ayars et al. [26] demonstrated that well-managed subsurface drip irrigation (SDI) systems can eliminate surface runoff, reduce deep percolation losses, and minimize surface water evaporation for annual crops. A recent study by Baram et al. [27] estimated leaching loss of 80–240 kg N/ha y from an almond (*Prunus dulcis*) orchard irrigated with micro-sprinklers, and most of the loss occurred early in the growing season (February–May), when urea–ammonium nitrate fertilizer was commonly applied to wet soil. Thus, high-frequency DI and fertigation could potentially reduce fertilizer losses and protect groundwater quality [17,28]. There has been no investigation of this possibility in pomegranate orchards.

A field experiment was conducted from 2010 to 2015 using high-frequency DI and subsurface (SDI) and fertigation to determine water and N requirements and yield response for pomegranate cv. Wonderful in the San Joaquin Valley, California [17]. Based on a lack of differences in yield among three levels of N application, the N requirement for a 4–6-year-old pomegranate orchard was concluded to be in the range of 62 to 112 kg/ha (109–198 g/tree), which is lower than what is used in current commercial practice. Although total yield was not significantly different between DI and SDI, SDI used less water by design, which resulted in less weed growth, and sometimes resulted in significantly higher either prime or subprime fruit yield compared to DI. Ayars et al. [17] did not examine N uptake data and how the irrigation systems and N treatment levels may have affected C and N dynamics. We hypothesized that the different drip irrigation systems significantly impact C and N distribution and transport in the soil profile, and that effective management practices could be developed with proper levels of N application for high yield and reduced environmental risks. The objective of this research

was to determine the effect of high-frequency DI and SDI and different levels of applied fertilizer N on C and N distribution in the soil, in pomegranate leaf tissue, and total N uptake.

## 2. Materials and Methods

### 2.1. Study Site

The study was conducted in a 1.4 ha pomegranate (*Punica granatum* L. cv. Wonderful) orchard at the University of California Kearney Agricultural Research and Extension Center (UC-KARE) located near Parlier, CA (36°36′3′′ N; 119°30′39′′ W; 103 m elevation). The soil at the experimental site is classified as Hanford sandy loam (coarse–loamy, mixed, superactive, non-acid, thermic Typic Xerorthents). The soil had a pH 7.5 (1:2 0.01 M CaCl$_2$); EC$_{25}$ (1:1) 171 µS cm$^{-1}$; and field capacity ~17%. The climate is classified as Mediterranean, characterized by mild wet winters and hot dry summers with low annual precipitation occurring primarily in winter. The average annual precipitation is about 200 mm, but annual potential evapotranspiration (ETo) is approximately 1350 mm. All agriculture in the region is irrigated. Young pomegranate trees about 0.91 m in height were planted in April 2010 with a 4.9 m row spacing and a 3.6 m tree spacing within a row (~567 trees/ha).

The experimental design was a split-plot, with two irrigation methods as main plot treatments and three N fertilization rates as the sub-plot treatments with five replicates. The main plot treatments were DI and SDI with the laterals installed on the surface or at a depth of approximately 55 cm, respectively. Both drip systems had two laterals per tree row, one on each side of the tree row at a distance of 1.1 m from the center of the tree line. The emitters applied water at 2 L/h and were spaced 0.46 m apart along the lateral. During the spring and summer of the first year (2010), while the SDI system was installed and the control pad was finished, all the trees were irrigated manually to ensure that they received enough water during the early growth stage. The same irrigation and N rates were applied to the entire orchard during the first two years (2010–2011) to ensure uniform plant establishment. The trees were around 2.44 m in height at the time N treatments started in 2012. Once the N treatments started, all the trees, including a lysimeter tree, were minimally pruned to maintain a bush-like shape and mechanically pruned at a height of 3.0 m [29]. Details about the field design and treatments can be found in Ayars et al. [17].

A large automated lysimeter (4 m long × 2 m wide × 2 m deep) containing one tree was used to manage the irrigation scheduling of the experimental site [17,30,31]. The lysimeter tree was irrigated using SDI treatment with the same number of emitters as the trees in the field. The trees in all N treatments were irrigated at 100% of crop water replacement as measured by the lysimeter. Three N rate treatments (N1, N2, and N3) started in 2012 and continued to the end of December 2015. In 2012, the trees received 52, 165, and 279 kg N/ha for N1, N2, and N3, respectively. As the trees grew larger, the N application rate increased. In 2013, the trees received 70, 166, and 245 kg N/ha for N1, N2, and N3, respectively. In 2014, the trees received 62, 223, and 342 kg N/ha for N1, N2, and N3, respectively. In 2015, the trees received 113, 263, and 331 kg N/ha for N1, N2, and N3, respectively. All N was applied through fertigation. N-pHURIC® 10/55 (urea and sulfuric acid with 10% N and 18% S) was applied to all levels of N treatments. The N-pHURIC® was used also to maintain the irrigation water at pH 6.5 to prevent precipitation of phosphates. For the additional N requirement in N2 and N3, ammonium nitrate (20% N or AN20) was injected. Phosphoric acid (H$_3$PO$_4$) and potassium thiosulfate (25% K, 17.5% S) were injected to all treatments at the same rate. The rate increased as trees grew larger. In 2013, all trees received 66 and 73 kg/ha of P and K, respectively. In 2014, all trees received 92 and 85 kg/ha of P and K, respectively. In 2015, all trees received 116 and 181 kg/ha of P and K, respectively. In 2012, when the differential N treatments were begun, an additional 10% water was applied to the DI treatments to compensate for potentially increased evaporative water loss and the weed growth that was controlled during the growing season. The trees were irrigated after 1 mm of crop water use had been measured in the lysimeter. During peak crop water use, this resulted in up to 8 to 10 mm per day. More details about the field and lysimeter operation are given in Ayars et al. [17].

## 2.2. Soil Measurements

Soil samples were collected twice a year (early spring before irrigation and fall after harvest but before the rainy season started) from 2012 to 2013 to determine total C, total N, and DOC. Samples were collected from all five treatment plots at eight soil depths (cm): 0–15, 15–30, 30–45, 45–60, 60–75, 75–90, 90–105, and 105–120 using a 7.6 cm diameter soil auger, oven dried at 65 °C for 48 h, ground with a Model 4-E Grinding Mill (QCG Systems, LLC, Phoenixville, PA, USA), and sieved through a 2 mm screen. Total C and total N contents were determined by dry combustion with a Flash 2000 N & C Soil Analyzer (Thermo Scientific®, Pittsburgh, PA, USA). The DOC and $NO_3^-$ concentrations were determined after mixing the soil with deionized water (1:1 soil:water) for 24 h, shaken for 1 h on a reciprocating shaker, and vacuum filtered through Whatman no. 42 filter paper. Carbon recovered in the water extract was determined using a Fusion Total Organic Carbon Analyzer TM (Teledyne Tekmar, Mason, OH, USA). $NO_3^-$ concentration was determined using an Astoria 2 Analyzer (Astoria Pacific Inc., Clackamas, OR, USA), and soil samples collected from 2012–2014 were analyzed.

## 2.3. Plant Measurements

Tree leaves were collected biweekly beginning in May 2012 and continuing to the end of September 2014 to determine effects of the irrigation system and N treatment on leaf N concentration as an indicator of tree N status [32]. Data for 2015 were reported in Ayars et al. [17]. The present paper presents the data up to 2014, when fruit samples were collected after harvest, to evaluate N distribution and uptake by the crop. Leaf samples were washed with deionized water, oven dried at 65 °C, and ground with an UDY Cyclone Sample Mill (UDY Corp., Fort Collins, CO, USA). Total N and C in the plant samples were measured using the same dry combustion method as for the soil samples. Since the changes in leaf concentrations during the growing season from May through September did not differ among treatments [17], the average values of all samples during this period of time each year were used for N uptake and C interpretation.

At harvest in October 2014, pomegranate fruit samples were collected from all treatments and replicates. The fruit were separated into arils and peels and were processed similarly to the leaves for analysis. Average fruit N concentrations were calculated based on weight ratio of arils to peels and were then used to estimate total N uptake in fruit by multiplying by the yield (yield data were reported by Ayars et al. [17]).

The MIXED procedure of SAS version 9.4 (SAS Institute, Cary, NC, USA) was used to fit a repeated measure mixed model. The fixed effects were irrigation type, N rate, soil depth, and their interactions. The random effects were replications (rep), irrigation × rep, and irrigation × N × rep. The latter interaction was used to define the experimental units for incorporating a first order, autoregressive covariance structure among the repeated measures in depth.

Focus was on the irrigation type and N rate effects, for which the least square means and their 95% confidence intervals were obtained. If TC was not detectable (ND), a zero value was used for analyses.

## 3. Results and Discussion

### 3.1. Soil Total C and N in Different Irrigation and N Treatments

Significant differences ($p < 0.05$) in soil TC concentration at the same depth between N treatments within irrigation type are shown in Table 1. In 2012, greater TC was observed in N3DI (0.3–0.8%) than in N1DI (<0.02%) and N2DI (0.02–0.04%) treatments in upper soil depths of 0–75 cm. However, greater TC was found in N3SDI (0.4–0.8%) than the other N treatments at 75–120 cm depths (<0.01%). TC in N1 and N2 treatments was low throughout the profile under both DI and SDI. At depths 75 cm the TC was ND in all DI treatments including N3DI, while in the SDI treatment, TC was significantly lower at 0–45 cm depth in the N3 treatment. The occurrence of high TC in the N3SDI treatment corresponded to the placement of the drip tubing (~50 cm) where fertilizer was delivered.

**Table 1.** Soil total C concentrations (%) in soil profile (0–120 cm) under different irrigation systems and N levels for 2012 and 2013.

| Soil Depth | Treatments (2012) | | | | | | Treatments (2013) | | | | | |
|---|---|---|---|---|---|---|---|---|---|---|---|---|
| | Drip Irrigation | | | Subsurface Drip Irrigation | | | Drip Irrigation | | | Subsurface Drip Irrigation | | |
| (cm) | N1 [z] | N2 | N3 | N1 | N2 | N3 | N1 | N2 | N3 | N1 | N2 | N3 |
| | (%) | | | | | | (%) | | | | | |
| 0–15 | 0.02 b [y] | 0.04 b | 0.82 a | 0.02 a | 0.08 a | 0.03 a | 1.11 b | 1.42 b | 3.73 a | 0.1b | 0.1 b | 1.03 a |
| 15–30 | ND b | 0.02 b | 0.78 a | 0.01 a | 0.03 a | 0.01 a | 1.15 b | 1.27 b | 3.26 a | 0.24 b | 0.15 b | 1.19 a |
| 30–45 | ND b | 0.02 b | 0.74 a | 0.01 a | 0.02 a | 0.01 a | 1.13 b | 1.18 b | 1.87 a | 0.36 b | 0.2 b | 1.19 a |
| 45–60 | ND b | 0.02 b | 0.46 a | 0.01 b | 0.03 b | 0.24 a | 1.1 a | 1.06 a | 1.06 a | 1.27 b | 1.02 b | 3.73 a |
| 60–75 | ND b | 0.02 b | 0.28 a | 0.01 b | 0.06 b | 0.76 a | 0.48 b | 0.9 a | 0.06 c | 0.9 b | 0.03 b | 2.38 a |
| 75–90 | ND | ND | ND | 0.01 a | 0.01 a | 0.68 a | 0.18 a | 0.11 a | 0.06 a | 0.48 b | 0.04 b | 2.1 a |
| 90–105 | ND | ND | ND | 0.01 b | ND b | 0.54 a | 0.1 a | 0.06 a | 0.02 a | 0.18 b | 0.04 b | 2.1 a |
| 105–120 | ND | ND | ND | ND b | ND b | 0.43 a | 0.03 a | 0.05 a | 0.02 a | 0.00 b | 0.03 b | 1.06 a |

[z] N1 = (N = 52–70 kg/ha in 2012–2013), N2 = (N = 165–166 kg/ha), N3 = (N = 245–279 kg/ha), and ND = non-detectable (zero percentage). [y] followed by the same letter between treatments for each soil depth within the year indicates no significant difference at $p < 0.05$.

In 2013, the TC percentage generally increased in all soil profiles in all treatments, up to 4× (~3.7%) times higher than in 2012 (Table 1). A similar distribution pattern of TC in 2013 to that in 2012 was observed. In the DI treatment, greater TC was found in the upper 60 cm of the soil in all levels of N treatments. TC decreased as soil depth increased and dropped to the ND at a 120 cm soil depth. The TC in N3DI was 2–3× that from the other two N treatments at 0–30 cm. In the SDI plots, TC was higher in N3 than all corresponding depths in N1 and N2 (average 1.5%), and was the highest at 45–60 cm in the profile.

There were significant effects of irrigation by N ($p < 0.05$) on TN in soil in 2012 and 2013. Total N distribution followed a similar pattern to the TC distribution with higher TN in the upper soil profile under DI and subsurface soil under SDI, and higher concentrations in N3 than N1 and N2 (Table 2). In 2012, higher concentrations of TN were found in the upper 45 cm of soil for N3DI compared with the remaining N treatments. The N3SDI soils had the highest TN at a 30–75 cm depth, which was greater than the other treatment combinations. At depths > 75 cm, all TN concentrations were low, and some were not detectable at 105–120 cm depths, indicating that N storage did not increase at this depth under high-frequency DI management. TN concentrations increased with increased depth in 2013 compared to 2012. Greater concentrations of TN were found in the upper 75 cm of soil in all N levels under DI with the highest under N3DI, but all dropped significantly with no differences among treatments below 75 cm soil depth. Under SDI, higher TN concentrations in 2013 were found only in N3SDI at a 75–90 cm depth compared with the other N treatments.

**Table 2.** Soil total N concentrations (%) in soil profile (0–120 cm) under different irrigation systems and N levels for 2012 and 2013.

| Soil Depth | Treatments (2012) | | | | | | Treatments (2013) | | | | | |
|---|---|---|---|---|---|---|---|---|---|---|---|---|
| | Drip Irrigation | | | Subsurface Drip Irrigation | | | Drip Irrigation | | | Subsurface Drip Irrigation | | |
| (cm) | N1 [z] | N2 | N3 | N1 | N2 | N3 | N1 | N2 | N3 | N1 | N2 | N3 |
| | (%) | | | | | | (%) | | | | | |
| 0–15 | 0.12 c [y] | 0.38 b | 0.82 a | 0.01 a | 0.11 a | 0.11 a | 0.27 b | 0.35 b | 0.91 a | 0.11 a | 0.06 a | 0.09 a |
| 15–30 | 0.11 c | 0.32 b | 0.76 a | 0.04 a | 0.09 a | 0.11 a | 0.26 b | 0.25 b | 0.72 a | 0.13 a | 0.08 a | 0.02 a |
| 30–45 | 0.12 c | 0.28 b | 0.37 a | 0.05 b | 0.1 b | 0.56 a | 0.23 b | 0.28 b | 0.54 a | 0.03 a | 0.08 a | 0.05 a |
| 45–60 | 0.15 a | 0.25 a | 0.22 a | 0.05 c | 0.19 b | 0.29 a | 0.12 b | 0.2 b | 0.54 a | ND a | 0.1 a | 0.02 a |
| 60–75 | 0.03 a | 0.11 a | 0.1 a | 0.08 b | 0.2 a | 0.25 a | 0.12 b | 0.1 b | 0.38 a | ND c | 0.11 b | 0.32 a |
| 75–90 | 0.03 a | 0.13 a | 0.07 a | 0.07 a | 0.22 a | 0.07 a | 0.06 a | 0.11 a | 0.02 a | ND b | 0.04 b | 0.27 a |
| 90–105 | ND a | 0.18 a | 0.07 a | 0.07 a | 0.2 a | 0.05 a | 0.04 a | 0.04 a | 0.02 a | ND b | 0.04 b | 0.15 a |
| 105–120 | ND a | 0.07 a | 0.06 a | 0.00 a | 0.21 a | 0.04 a | 0.02 a | 0.04 a | 0.00 a | ND a | 0.03 a | 0.00 a |

[z] N1 = (N = 52–70 kg/ha in 2012–2013), N2 = (N = 165–166 kg/ha), N3 = (N = 245–279 kg/ha), and ND = non-detectable (zero percentage). [y] followed by the same letter between treatments for each soil depth within the year indicates no significant difference at $p < 0.05$.

In our study, the total C in soil was low (<1%) in 2012. In 2013, total C increased up to ~3.5% in soils treated with a higher N application rate (N3) in the upper profile of the soil (0–30 cm) under DI, and ~3.0% in the subsurface profile (45–75 cm) under SDI. Higher TC with DI could have been attributed to weed growth (Ayars et al. [17]). Both TC and TN increases followed a similar pattern, i.e., in surface soil under DI and subsurface soil near the drip line under SDI. The higher TC values associated with higher N application rate indicated that N supply under adequate soil moisture must have benefited the production of microbial biomass. Decock et al. [33] found similar results in almond orchards in the upper 15 cm of well-drained, gravelly sandy loam soils (Arbuckle series—Typic Haploxeralfs) in California, which were irrigated with a microjet sprinkler system twice a week for a total of 380 mm/ha y and fertilized with N (urea ammonium nitrate, 32% N) three to five times a year up to 258–280 kg N/ha y. They found that TC, TN, and C/N ratios were approximately 0.83%, 0.083%, and 10, respectively. The increase in TC with time may have been explained by Wolf et al. [34], who reported that most crop systems experienced a small increase in soil C from both conventional and organic productions for annual crops. The larger TC (an average of ~3.5%) increase in our study with high-frequency DI in 2013 in the first 30 cm of the soil could indicate some advantage of DI in orchards that may promote soil organic carbon (OC) build-up to improve soil productivity, which warrants further in-depth investigation. Examination of soil total organic C to N ratio (Figure 1) also indicated a very low ratio of 1:1 under DI, an undefinable ratio under SDI in 2012, and a C:N ratio increase to about 3.7:1 under DI and to 4.2:1 under SDI in 2013. In apple orchards in temperate climates, the C:N ratio ranged from 6.1–9.5:1 [35] and other reported values ranged from 10:1 to 19:1 [36]. As both soil OC and N play an important role in soil sustainability, a C:N ratio of 24:1 has been suggested to maintain soil health [37]. Under these optimum conditions, soil microbes can spur the release of nutrients like N, P, and Zn to crops while maintaining the necessary amount of soil-protecting residue. The lower C:N ratio in our study was caused a result of low soil organic matter (<1%) that suggests a need to increase soil OC to sustain or increase soil productivity in the region. The relatively higher C:N ratio under SDI may indicate some advantage over DI, which requires further investigation.

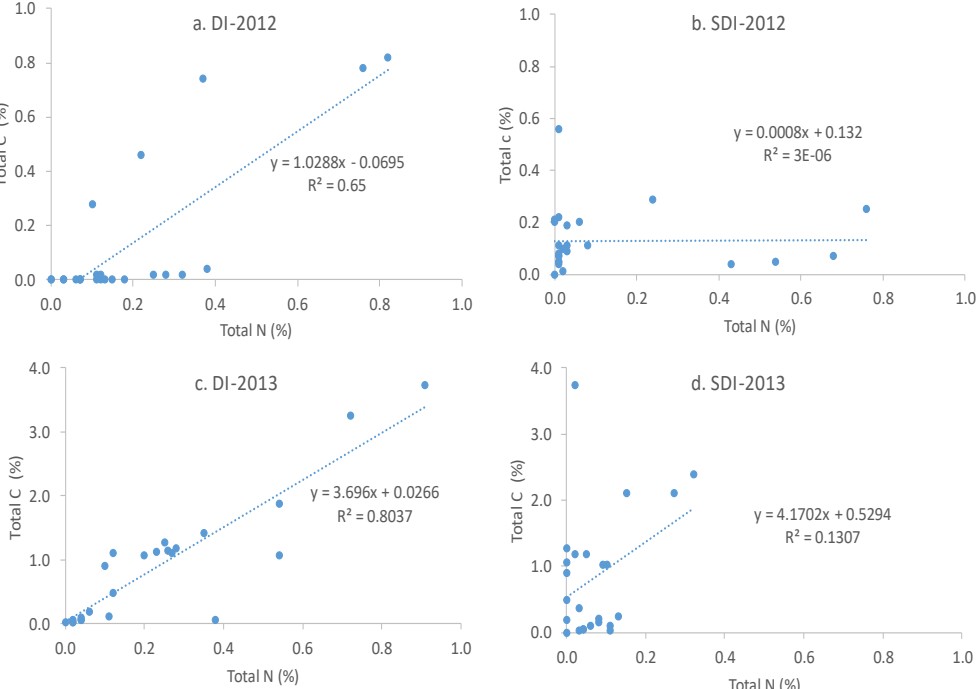

**Figure 1.** Correlation between soil total carbon and nitrogen under different irrigation systems (DI = drip irrigation and SDI = subsurface irrigation) for years 2012 and 2013.

### 3.2. Soil DOC and NO₃⁻ in Different Irrigation and N Treatments

In both 2012 and 2013, DOC concentrations and distributions in soil were significantly affected by the irrigation type × N rate interaction ($p < 0.05$). In 2012, under DI, a greater DOC was found in the upper 45 cm soil for N3 than N2, which had a significantly higher DOC than in N1 (Figure 2). There was no statistical difference for soil depths below 45 cm. Under SDI, however, in addition to lower values in surface soils compared to those under DI, DOC concentration and distribution were all similar with no differences among the three levels of N treatments, which was likely due to a relatively dry surface with N applied at a greater depth. In 2013, greater DOC was found for N3 treatments than for N2 and N1 under both DI and SDI, with significantly higher concentrations in the upper 75 cm under DI and at ≤40 cm depth under SDI.

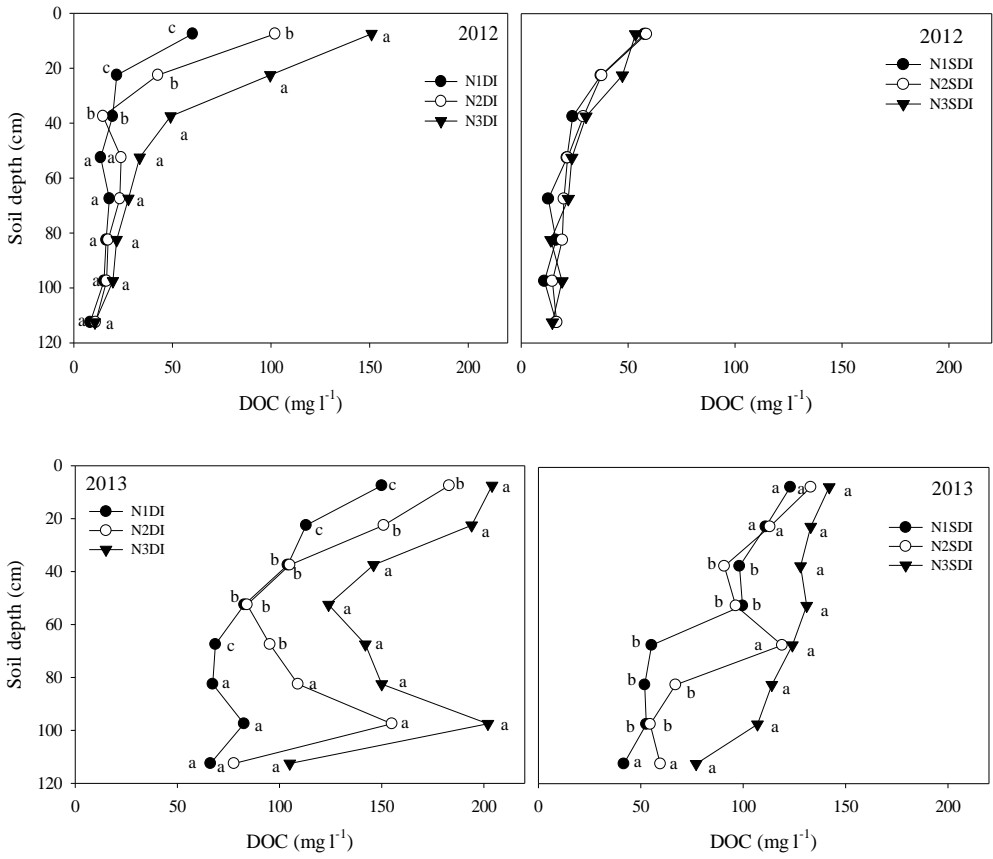

**Figure 2.** Concentrations of dissolved organic carbon (DOC) in a 1:1 mass soil extract using distilled water from soil samples collected under different irrigation systems and nitrogen levels for 2012–2013. Treatments: DI = drip irrigation, SDI = subsurface irrigation, N1 = (N = 52–70 kg/ha in 2012–2013), N2 = (N = 165–166 kg/ha), and N3 = (N = 245–279 kg/ha). Means followed by the same letter or no letters between treatments for each soil depth within the year were not significantly different at $p < 0.05$.

For both 2012 and 2013, DOC concentration increased significantly from N1 to N2 to N3 under DI, and DOC concentrations from the N3 treatment under SDI were significantly higher than those from N1 and N2 in 2013 only. As DOC are the results of decomposition processes from organic materials, the data suggest that increased N level increased decomposition of soil organic materials. The effects of N on the decomposition of organic materials have been examined in several studies, including meta-analyses in recent years [38–41]. Nitrogen addition significantly impacted litter decomposition rate in Songnen meadow ecosystems [42]. Nitrogen fertilizer may increase the loss of cellulose, but it suppresses the breakdown of lignin in plant litter [43]. Similar conclusions were derived from meta-analyses indicating that N addition reduced the activity of lignin-modifying enzymes (LMEs), and this N-induced enzyme suppression was associated with increased soil organic C [39]. Jian et

al. [40] showed that N fertilization significantly increased some enzyme activities while decreasing others after analyzing data from 65 published studies. Nitrogen fertilization generally enhanced SOC and TN but inhibited soil microbial biomass carbon. The contrasting effects of N addition on cellulose (hydrolytic C-degrading enzymes) and ligninase (oxidative C-degrading enzymes) activities were summarized by Luo et al. [41], who concluded that N-enhanced cellulose activity contributed to a minor stimulation of soil respiration, whereas N-induced repression on ligninase activities led to soil C sequestration. The net results were increased DOC with increased N level treatments in our field study.

Nitrate distribution in soil was also significantly affected by irrigation and N rate interactions ($p < 0.05$). In 2012, higher $NO_3^-$ was found under N3 compared with the other two levels of N under DI in the upper 60 cm of soil (Figure 3). Under SDI, higher $NO_3^-$ concentrations were observed for N3 versus the other two N levels at 30–60 cm soil depths. The generally higher concentrations in SDI treatments or no difference from the DI treatments in the upper profile may be attributed to the carry-over from the previous year, as no N was applied to the surface in 2011. In 2013, higher $NO_3^-$ was found under N3 and N2 compared with N1 under DI in the upper 30 cm soil and also at greater depths (90–120 cm). The increased $NO_3^-$ concentration at greater soil depths may indicate that leaching occurred in N3 treatments. For SDI, greater $NO_3^-$ concentrations were found at 45–75 cm soil depth from N3 and N2 compared with N1. As $NO_3^-$ is highly mobile in soil, the much higher concentrations at 30–40 cm depths might reflect a temporary increase from the downward movement of surface $NO_3^-$ in early spring, which is supported by the reduced concentration from 2012 to 2013 in the upper 20 cm of soils. In 2014, higher $NO_3^-$ was again found below 80 cm soil under DI from N3 and N2 compared with N1, while under SDI, greater $NO_3^-$ concentration was determined in subsurface soil at 30–75 cm depths from N3 versus the other two N levels. These data indicate that higher leaching could have occurred due to excessive N from N3 than N1 and N2 treatments. Excluding N3 treatments, the soil profile data showed more N storage in DI than SDI, which could be partially due to higher N plant uptake from SDI. In any case, the higher storage of $NO_3^-$ under DI means higher leaching loss when the rainy season arrives.

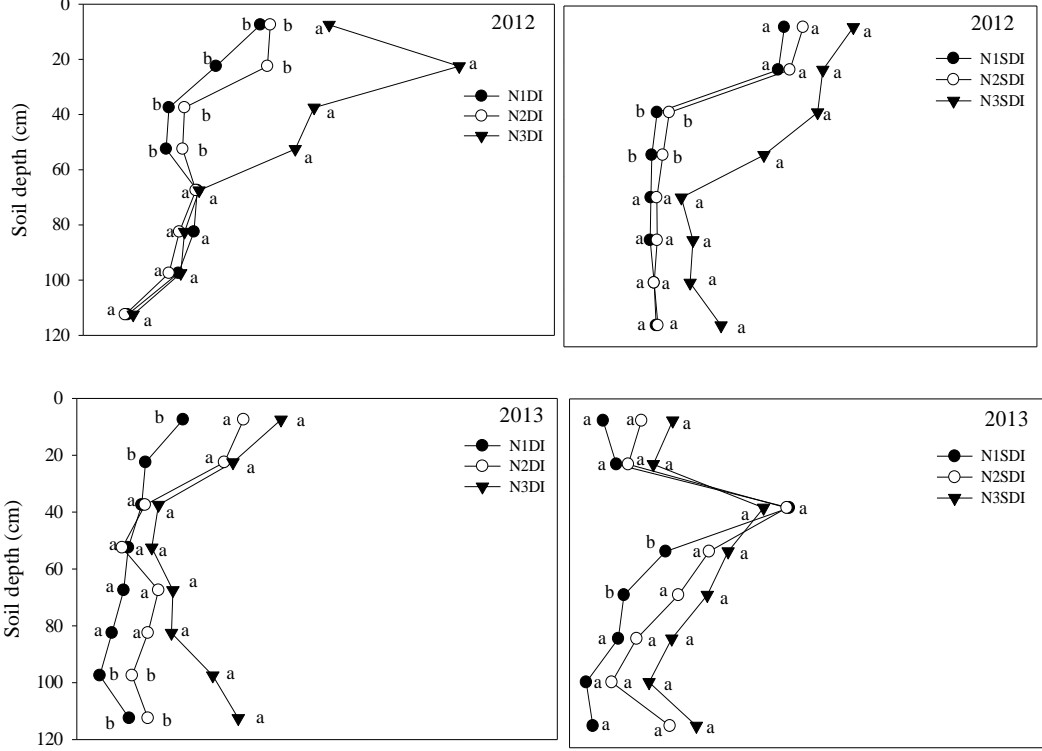

**Figure 3.** *Cont.*

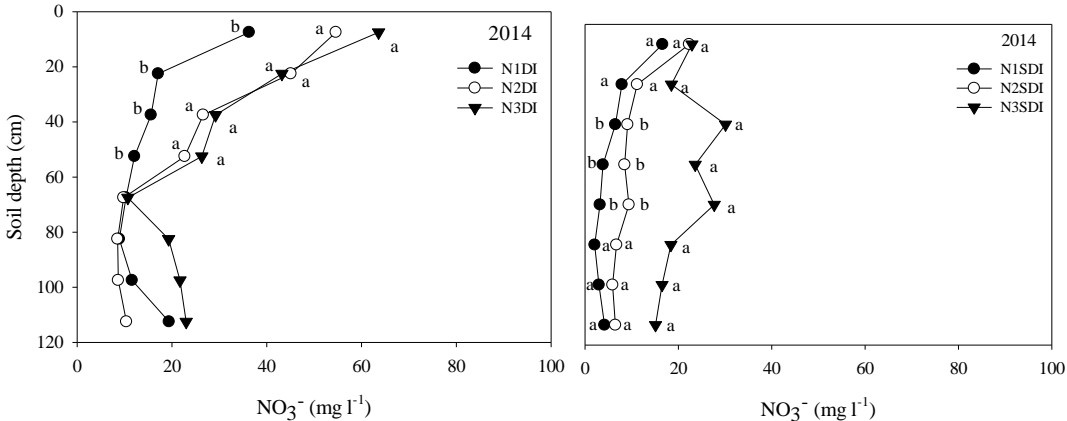

**Figure 3.** Concentrations of nitrates ($NO_3^-$) in a 1:1 mass soil extract using distilled water from soil samples collected under different irrigation systems and nitrogen levels for 2012–2014. Treatments: DI = drip irrigation, SDI = subsurface irrigation, N1 = (N = 52–70 kg/ha in 2012–2013), N2 = (N = 165–166 kg/ha), and N3 = (N = 245–279 kg/ha). Means followed by the same letter or no letters between treatments for each soil depth within the year were not significantly different at $p < 0.05$.

Our data revealed the significant impact of irrigation type and N level on soil C and N mobility. DOC and $NO_3^-$ availability and movement through the soil profile are influenced by not only soil texture and local climate, but also by agricultural practices such as irrigation type, application of different rates of N, fertigation practices, and vegetation cover versus bare soil [38]. DOC concentrations can be used as an indicator of substrate availability of readily available C and $NO_3^-$ concentrations for denitrification [33]. Gu et al. [44] found higher DOC (~23%) in the first 40 cm of soil of an established citrus orchard growing in a sandy loam soil covered with grass versus bare soil. They suggested that higher DOC could be responsible for the growth of vegetation, reduction of soil temperatures, and evaporation of water. In our study, we found that higher DOC was associated with higher N application and increased with crop years (i.e., from 2012 to 2013). Over time, leaves falling from these deciduous trees may lead to increased surface soil organic materials. Also observed in 2012 was a 2 to 3× higher DOC in surface soil (up to 45 cm of soil depth) under the DI system compared with SDI, which was partially attributed to weed growth. Ayars et al. [17] reported weed biomass 7× higher under a DI (589 g/m²) versus a SDI (81 g/m²) system, and 1.5× higher from N3 or N2 (~365 g/m²) than N1 (~250 g/m²). Decock et al. [33] found that in almond orchards growing in sandy loam soils in California irrigated with microjet sprinkler system and fertilized with N up to 258–280 kg N/ha y, $NO_3^-$ concentrations typically increased following fertilizer application. The $NO_3^-$ and DOC concentrations in the top 15 cm of the soil ranged between 0–128 mg/L and 0–310 mg/L, respectively, from samples collected in the tree row.

*3.3. Total N Concentration and C/N Ratio in Leaves*

Total N in tree leaves was analyzed multiple times in each year and was used to characterize the response to the N treatments. The main effects are shown in Table 3, where no significant effect was found for the irrigation by N rate interaction ($p > 0.05$). TC exhibited similar values ranging from 48%–56% for years 2012–2014. The TN and C/N ratios were only significantly affected by the main effects. Higher TN was found in DI versus SDI treatment in 2012 only. Additionally, about 5% higher TN was found in N2 and N3 compared with N1 trees in 2013. For the C/N ratio, a higher value was found in SDI versus DI in 2012. In addition, a higher C/N ratio was found in N1 versus N2 or N3 treatments in 2012 and 2014. Based on the results, there were no significant differences between N2 and N3, indicating that N2 application was sufficient to meet pomegranate tree demands for maximum uptake. Sebastian et al. [45] reported that in mature almond orchards growing in California, leaf N concentration was significantly higher from >224 kg N/ha application rates than from 140 kg N/ha.

**Table 3.** Total N and C/N ratio of pomegranate leaf tissue samples treated with three levels of N under two irrigation systems.

| Year | Total N (%) | | | | | C/N Ratio | | | | |
|------|------|------|------|------|------|------|------|------|------|------|
| | Irrigation | | N Rate | | | Irrigation | | N Rate | | |
| | DI [z] | SDI | N1 [y] | N2 | N3 | DI | SDI | N1 | N2 | N3 |
| 2012 | 1.70 a [x] | 1.65 b | 1.64 | 1.69 | 1.70 | 32.1 b | 32.4 a | 32.8 a | 31.3 b | 31.8 b |
| 2013 | 1.94 a | 1.93 a | 1.88 b | 1.95 a | 1.96 a | 28.1 a | 28.2 a | 28.2 a | 28.3 a | 28.4 a |
| 2014 | 1.96 a | 1.94 a | 1.88 a | 1.96 a | 2.01 a | 24.9 a | 25.1 a | 26.0 a | 24.7 b | 24.4 b |

[z] DI = drip irrigation and SDI = subsurface irrigation. [y] N1 = (52–70 kg/ha in 2012–2013), N2 = (165–166 kg/ha) and N3 = (245–279 kg/ha). [x] Means followed by the same letter between treatments (i.e., irrigation system or N rate) for each soil depth within the year were not significantly different at $p < 0.05$.

N is highly mobile in plants, which contributed to the distribution pattern. Ayars et al. [17] reported that leaf N contents are significantly affected by the N rate and irrigation type × N interactions, with reduced effects on N in fruit peels or arils. Although no biomass data were collected to estimate TN uptake by the trees, the lowest N concentrations in tree fruits from N1 (the only treatment with lower N concentration in fruits than branches) may suggest that the N supply from N1 was not sufficient to meet tree growth demands. However, results from Zhang et al. [29] demonstrated very little difference in canopy size due to irrigation types and N application. The lack of differences in the concentrations between N2 and N3 in the present study supported the conclusion that the N2 treatment provided sufficient N and the higher application at N3 was excessive for pomegranate tree growth.

*3.4. Total N Uptake by Fruit*

Data on total N uptake in fruit (N concentration in fruit × yield) are shown in Table 4. The N uptake was greater (by 10%) with SDI treatment than with DI in 2014 and 2015. Meanwhile, for N treatments, a higher fruit N uptake occurred in the N2 and N3 treatments in 2013 and 2014, but not in 2015 when compared with N1. There were no significant differences ($p > 0.05$) in the N uptake between N2 and N3 treatments in any year. The data suggest that SDI resulted in a ~1.1× higher N use efficiency than DI. However, the higher N application rate for N3 did not result in higher N uptake in comparison with the N2 rate. The data further support that the N2 rate (120, 138, and 159 kg N/ha for years 2013, 2014, and 2015, respectively) provided sufficient N for the crop.

**Table 4.** Fruit N uptake for 2013, 2014, and 2015 by irrigation systems and N levels.

| Year | Irrigation System | Fruit kg N/ha | N Treatment | Fruit kg N/ha |
|------|------|------|------|------|
| 2013 | | | | |
| | DI [z] | 105 a | N1 [y] | 95 b |
| | SDI | 113 a | N2 | 120 a |
| | | | N3 | 112 a |
| 2014 | | | | |
| | DI | 119 b [x] | N1 | 108 b |
| | SDI | 135 a | N2 | 138 a |
| | | | N3 | 135 a |
| 2015 | | | | |
| | DI | 141 b | N1 | 127 a |
| | SDI | 154 a | N2 | 159 a |
| | | | N3 | 156 a |

[z] DI = drip irrigation, SDI = subsurface irrigation. [y] N1 = (52–70 kg/ha in 2012–2013), N2 = (165–166 kg/ha) and N3 = (245–279 kg/ha). [x] Means followed by the same letter between treatments (i.e., irrigation system or N treatment) for each year were not significantly different at $p < 0.05$.

The total N uptake by fruit included both prime and subprime fruit. Prime fruit were defined as fruit with good color, greater than 8 cm diameter, with no visible cracks, and marketable as fresh fruit. Subprime fruit were those suitable for juicing and contained green and open-cracked fruits. Ayars et al. [17] evaluated treatment effects on total fruit yield and also prime and subprime fruit yield data (2013–2015). They found no statistical differences in yield between irrigation methods or N treatments, nor were there interactions on total yield, although there were some statistical differences in the prime and subprime fruits; thus, they concluded that N1 provided the N requirement, but our data presented in Table 4 showed that the N2 rate (application of 166–263 kg N/ha from 2013–2015) resulted in greater N uptake in fruit, leading us to conclude that the N2 rate provided sufficient N for tree growth and yield. Since all fruit were removed at harvest, the total N removed from the orchard by harvest should be used in N management, and the TN uptake data clearly showed effects from both irrigation and N application levels.

Additionally, a higher N application rate (N3) was excessive because there are no differences in total N uptake or yield (Ayars et al. [17]) compared with N2 treatment (Table 4). The total N uptake by fruit was 62% and 40% of total N applied for the N2 (223 kg/ha) and N3 (342 kg/ha) treatments, respectively, in 2014. Considering N storage in tree trunks and branches, the total N use efficiency was >62% for N2, which is considered high in tree crops. Thus, the N2 rate (263 kg/ha applied in 2015) provided sufficient N for the trees. Similar results were also reported by Muhammad et al. [23], who found a positive effect of N rate in nut N uptake in mature almond orchards growing in well-drained Milham sandy loam soils (Typic Haplargids) in California, irrigated with a micro-sprinkler system and fertilized with four N treatments (140, 224, 309, and 392 kg/ha). For four years (2008–2011), the highest N uptake was observed from the 309 kg/ha application rate.

## 4. Conclusions

This study evaluated the effect of DI versus SDI and three N application levels on TN, TC, $NO_3^-$, and DOC distribution in the soil as well as TN concentration or distribution in pomegranate tree tissues and total fruit N uptake in a fully irrigated orchard. The different irrigation designs resulted in very different C and N distributions in the soil profile. DI resulted in higher concentrations of TN, TC, $NO_3^-$, and DOC in soil depths above 75 cm compared with SDI, which resulted in higher concentrations in soil depths below 30 cm. This was attributed to the placement of drip tubing at ~50 cm soil depth in the SDI treatment. Higher TC and DOC in DI irrigation versus SDI can be attributed to a higher presence of weeds and tree growth over time, which contributed to generally increased surface soil OC. Overall data from soil analyses indicated that the N3 rate appeared to be excessive, with no yield increase, although it did increase the concentration of TN, TC, $NO_3^-$, and DOC in deep soils under both DI and SDI, indicating an increased leaching risk that must be avoided. Furthermore, the plant data indicated lower N utilization in N3 than N2. At or below the N2 (166–263 kg/ha for 4–6 year old pomegranate trees) application rate, high-frequency SDI showed no increased leaching risk compared to DI. The results indicate that a high-frequency irrigation method with proper N application rate could be managed to achieve high yield or high N uptake while reducing chemical input and leaching loss. Based on fruit N uptake, the N2 application rate was sufficient to meet maximum pomegranate tree N requirement. High-frequency SDI is an efficient system that can be recommended to San Joaquin Valley pomegranate farmers to save water, increase N uptake, and reduce leaching loss to protect groundwater quality.

**Author Contributions:** R.T.-C. and S.G. run the statistical analysis of the different measured variables and R.T.-C. and S.G. wrote the manuscript. J.E.A., C.J.P., D.W., and R.C.P. experiment logistic. All the authors revised the manuscript.

**Funding:** This research received no external funding.

**Acknowledgments:** We acknowledge California Department of Food and Agriculture (CDFA) for supplemental funding, KARE Center for the field site, and the following companies for contributions to this project: Paramount Farming—trees and cones, Toro Irrigation—drip tubing, Lakos Filtration—sand media, Dorot—electronic control

valves, Verdegaal Brothers—fertilizers, and SDI+—consulting time and miscellaneous equipment. In addition, the authors would like to thank Rick Schoneman for his assistance with the irrigation and fertigation systems. Also, we would like to thank Aileen Hendratna, Ryan Lancaster, Don Tucker, Matthew Gonzalez, Julianne Anaya, and Phyllis Ukatu for helping in the collection, preparation, and analysis of soil and plant samples at the San Joaquin Valley Agricultural Science Center, USDA-ARS-WMRU in Parlier, CA. In memory of Donald J. Makus, who helped us with pomegranate yield collection and analysis.

**Conflicts of Interest:** The authors declare no conflict of interest.

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
