# Peer review of "Carbon and Nitrogen Dynamics Affected by Drip Irrigation Methods and Fertilization Practices in a Pomegranate Orchard"

_horticulturae, doi:10.3390/horticulturae5040077_

Round 1
Reviewer 1 Report
Tirado-Corbalá et al., investigated the effects of irrigation and N fertilization on C and N distribution in plant and soil. Their results are interesting and useful. Considering the amount of data, I would recommend major revision. The authors need better data interpretation.
Abstract
Overall, too many abbreviations in the abstract. I have to remember them when I read your abstract.
Line 22. What is the rate for N fertilization?
You need a better logical for your key results in the abstract, so that readers can easily remember your findings.
You only have implications for N in the last sentence, but how about C.
Introduction and discussion
The key questions and hypotheses are not such clear.
It seems that you have a lot of data, but you also need some good logic to persuade others that your results make some contribution to our understanding of some implications for the application.
There are some meta-analyses on the effects of nitrogen addition on soil carbon and nitrogen degrading enzymes, cellulase, and ligninase, which might be useful for your revision.
Data analysis
You need better data interpretation for your results. I think these results are very interesting and timely and very important. You need some substantial efforts to look behind the data.
Author Response
Reviewer #1
Abstract
Overall, too many abbreviations in the abstract. I have to remember them when I read your abstract.
I have removed some of the abbreviations to make the abstract clearer and easy to remember. (e.g., Lines 20, 31, 34).
Line 22. What is the rate for N fertilization?
The information has been added to the abstract. Three N rates (N1-50, N2-100 and N3-150% of current practice, respectively). As tress grew larger, N application rate increased. From 2013-2015, trees received the following rates of N: 62- 113 (N1), 166-263 (N2) and 244-342 kg/ha (N3). (Lines 23-25).
You need a better logical for your key results in the abstract, so that readers can easily remember your findings.
The abstract has been revised for easier read. (See revised Abstract)
You only have implications for N in the last sentence, but how about C.
The information has been added. (Line 34-35).
Introduction and discussion
The key questions and hypotheses are not such clear.
The introduction was revised with more text added to explain the importance of this research. Lines 41-43, 46, 54-56, 67-68, 85, 95-98.
It seems that you have a lot of data, but you also need some good logic to persuade others that your results make some contribution to our understanding of some implications for the application.
We made modifications in the Results and Discussion as well as Conclusions in the efforts to clearly state new information or the contribution of this research to our understanding how irrigation and N application level impact dynamics or C and N. (e.g., Lines 293-311, 373-392).
There are some meta-analyses on the effects of nitrogen addition on soil carbon and nitrogen degrading enzymes, cellulase, and ligninase, which might be useful for your revision.
Yes. After reviewing more literature, we have now added new information to explain data on soil DOC that was significantly affected by N fertilization level. See Lines 255-273. We do feel that this addition has improved the paper by advancing our knowledge on N and C dynamics in soil systems. Again, we appreciate the reviewer’s suggestions. See additions in Lines 255-273.
For both 2012 and 2013, DOC concentration increased significantly from N1 to N2 to N3 under DI and under SDI, DOC concentration from N3 treatment were significantly higher than those from N1 and N2 treatments in 2013. As DOC are the results of decomposition processes from organic materials, the data suggest that increased N level had increased decomposition of soil organic materials. The effects of N on decomposition of organic materials have been examined in several studies including those using meta-analysis in recent years [39-41]. Nitrogen addition was found to significantly impact litter decomposition rate in the Songnen meadow ecosystems [42]. Nitrogen fertilizer often increases the loss of cellulose but it suppresses the breakdown of lignin in plant litter [43]. Similar conclusion was derived from meta-analysis that N addition reduced the activity of lignin-modifying enzymes (LMEs) and this N-induced enzyme suppression was associated with increases in soil organic C [39]. Jian et al. [40] showed the N fertilization significantly increased some enzyme activities while decreased others after synthesizing data from 65 published studies. Nitrogen fertilization generally enhanced SOC and TN but inhibited soil microbial biomass carbon. The contrasting effects of N addition has contrasting effects on cellulose (hydrolytic C-degrading enzymes) and ligninase (oxidative C-degrading enzymes) activities was summarized by Luo et al. [41] that N-enhanced cellulose activity contributes to the minor stimulation of soil respiration whereas N-induced repression on ligninase activities leads to soil C sequestration. The net results were the increased DOC from increased N level treatments in our field study.
Data analysis
You need better data interpretation for your results. I think these results are very interesting and timely and very important. You need some substantial efforts to look behind the data.
See various additions in Results and Discussion.
Reviewer 2 Report
The topic is within the general scope of the journal. It is original but lacks relevance.
It is not really demonstrate the pertinence of the study. The study is not a long term, and characteristics measured (soil nitrate leaching and DOC) and plant N, especially pomegranate fruits different in time periods. Besides, statistical treatment is not fully developed. Some data are commented as they have been treated statistically which was not true.
There are several minor issues along the text, as follows:
Keywords; delete the last one and replace for a new one not included in title; use the singular form.
M&M: please, clarify the experimental layout and explain the reason for using different types of N fertilizers. This is not convincing.
Justify convincingly the need for analysing leaves during one year, and for analysing differently parts of stems (data not shown).
L196/198: please indicate the values
L208: "4%": what about the CV?
-L219: "4%": is this a coincidence?
-L223/224: which soil depth?
Section 3.2 is not convincingly addressed.
-L290/301, L322/330: discussion needs an improvement. Please shows the novelty.
-L304: "long term": this is not a long term experiment
-L362: do you mean, annual application? This is too high! You need to take into account the internal N reserves.
Author Response
R2-Open Review
The topic is within the general scope of the journal. It is original but lacks relevance.
It is not really demonstrating the pertinence of the study. The study is not a long term, and characteristics measured (soil nitrate leaching and DOC) and plant N, especially pomegranate fruits different in time periods. Besides, statistical treatment is not fully developed. Some data are commented as they have been treated statistically which was not true.
We have removed the questionable data from the text and added more relevant information (Lines 330-341.
There are several minor issues along the text, as follows:
Keywords; delete the last one and replace for a new one not included in title; use the singular form.
Deleted the last keyword “Pomegranate” and added “N uptake”.
M&M: please, clarify the experimental layout and explain the reason for using different types of N fertilizers. This is not convincing.
The experimental layout was provided in detail in Ayars et al. (2017). To avoid repetition, we cited this reference (Line 131) for readers to obtain more detailed information if needed. N-pHURIC® 10/55 (Urea and sulfuric acid with 10% N and 18% S) was used for all treatments first because the irrigation water was hard water and this fertilizer is used to reduce pH and avoid clogging problems. N-pHURIC® is more expensive (we do not have exact costs) and AN20, which was used for make-up N fertilizers for N2 and N3. These were made clear in Lines 140-143.
Justify convincingly the need for analyzing leaves during one year, and for analyzing differently parts of stems (data not shown).
Leaf N concentration has been used popularly to indicate N status in plants. We added a reference and statement for this (Line 163), plus it is much easier to sample. We decided to delete branch sample data because we do not have enough replications for statistical analyses.
L196/198: please indicate the values.
I have included the values in Lines 194-200.
L208: "4%": what about the CV?
Instead of using round numbers, I used the actual values presented in Table 1 entitled: “Soil total C concentrations in soil profile (0-120 cm) under different irrigation systems and N levels for 2012-2013”. Below are in red the actual values. Just make sure those values are the average numbers from 0-15 and 15-30 cm soil depth intervals in DI and 45-60 and 60-75 cm in SDI. In our study, the total C in soil was low (<1%) in 2012. In 2013, total C increased up to ~ 3.5 % in soils treated with higher N application rate (N3) in surface soil (0- 30 cm) under DI and ~ 3.0 % in subsurface (45-75 cm) under SDI.
-L219: "4%": is this a coincidence?
Is not a coincidence. As I mentioned in the previous comment I round the numbers in the script. This time I used the actual number to avoid confusion or create doubt.
-L223/224: which soil depth?
I now have included the soil depth and more relevant information. Decock et al. [33] found alike results in almond orchards growing in the first 15 cm of soil of well-drained, gravelly sandy loam soils (Arbuckle series -Typic Haploxeralfs) in California, which were irrigated with microjet sprinkler system twice a week for a total of 380 mm/ha yr and fertilized with N (urea ammonium nitrate, 32% N) three to five times a year up to 258–280 kg N /ha yr.
Section 3.2 is not convincingly addressed.
-L290/301, L322/330: discussion needs an improvement. Please shows the novelty. We add more information and make it more concise.
-L304: "long term":
this is not a long term experiment. I removed long term from this part of the script.
-L362: do you mean, annual application? This is too high! You need to take into account the internal N reserves. done
Round 2
Reviewer 1 Report
Publishable now.
Author Response
Dear Reviewer, thanks for the time you spend in the first version of the manuscript.Have a nice day, Rebecca